



# Prediction of seismic p-wave velocity using machine learning

Ines Dumke[1], Christian Berndt[1]

[1]GEOMAR Helmholtz Centre for Ocean Research Kiel, Kiel, Germany

*Correspondence to*: Christian Berndt (cberndt@geomar.de)

**Abstract.** Measurements of seismic velocity as a function of depth are generally restricted to borehole locations and are therefore sparse in the world's oceans. Consequently, in the absence of measurements or suitable seismic data, studies requiring knowledge of seismic velocities often obtain these from simple empirical relationships. However, empirically derived velocities may be inaccurate, as they are typically limited to certain geological settings, and other parameters potentially influencing seismic velocities, such as depth to basement, crustal age, or heatflow, are not taken into account. Here, we present

a machine learning approach to predict seismic p-wave velocity ($v_p$) as a function of depth ($z$) for any marine location. Based on a training dataset consisting of $v_p(z)$ data from 333 boreholes and 38 geological and spatial predictors obtained from publically available global datasets, a prediction model was created using the Random Forests method. In 60 % of the tested locations, the predicted seismic velocities were superior to those calculated empirically. The results indicate a promising potential for global prediction of $v_p(z)$ data, which will allow improving geophysical models in areas lacking first-hand velocity

data.

## 1 Introduction

Seismic p-wave velocities ($v_p$) and velocity-depth profiles are needed in many marine-geophysical applications, e.g. for seismic data processing, for time-depth conversions, or to estimate hydrate concentrations in gas hydrate modelling. Direct measurements of seismic velocities, however, are sparse and limited to borehole locations such as those drilled by the Deep

Sea Drilling Project (DSDP), the Ocean Drilling Program (ODP), and the International Ocean Discovery Program (IODP). Seismic velocities can also be obtained indirectly from seismic data. Approaches include derivation of 1D velocity profiles via refraction seismology using ocean bottom seismometers (OBS) (Bünz et al., 2005; Mienert et al., 2005; Westbrook et al., 2008; Plaza-Faverola et al. 2010a, 2010b, 2014), and velocity analysis of large-offset reflection seismic data (Crutchley et al., 2010, 2014; Plaza-Faverola et al., 2012). However, suitable seismic datasets are only available in certain areas, and OBS-derived

velocity profiles are of relatively low spatial and vertical resolution.

In the absence of measurements and refraction seismic data, constant velocities are often used for time-depth conversions (e.g. Brune et al., 2010) or processing of reflection seismic data (Crutchley et al., 2010, 2011, 2013; Netzeband et al., 2010; Krabbenhoeft et al., 2013; Dumke et al., 2014), even though a constant velocity-depth profile is generally unrealistic and will thus lead to inaccurate results.



As an alternative, empirical velocity functions have been derived, which are based on averaged measurements and provide seismic velocity-depth relationships for different geological and geographical settings. For example, Hamilton (1979, 1980, 1985) used averaged $v_p$ measurements from boreholes and sonobuoys to derive velocity-depth functions for different marine settings and sediment types. Velocities calculated from these empirical functions have been used e.g. for time-depth

conversions (Lilly et al., 1993; Brune et al., 2010), brute stack processing of reflection seismic data, as well as local (Bünz et al., 2005) and regional (Scanlon et al., 1996; Wang et al., 2014) velocity models.

Although velocity profiles calculated from empirical functions may work well in some cases, empirical functions do not always produce accurate $v_p(z)$ profiles, due to their use of depth as the only input parameter and their limitation to certain regions or geological settings. Mienert et al. (2005) observed both agreements and disagreements between velocity profiles derived from

OBS data and calculated from Hamilton functions, whereas Westbrook et al. (2008) argue that empirical functions are in general not representative for other areas due to variations in lithology and compaction history. Moreover, the Hamilton functions fail to provide correct velocities in areas containing gas hydrates or gas-saturated sediments (Bünz et al., 2005; Westbrook et al., 2008). Consequently, an alternative method is required to estimate $v_p(z)$ profiles for a larger variety of geological settings.

Over the last years, parameters in many different applications have been successfully predicted using machine learning techniques (e.g. Lary et al., 2016). In geosciences and remote sensing, machine learning methods have been used to predict soil properties (Gasch et al., 2015; Ließ et al., 2016; Meyer et al., 2018), air temperatures (Meyer et al., 2016a, 2018), biomass (Meyer et al., 2017), and the elasticity modulus of granitic rocks (Karakus, 2011). Applications also extended into marine settings, involving the prediction of sediment mud content off southwest Australia (Li et al., 2011), as well as parameters such

as seafloor porosity (Martin et al., 2015; Wood et al., 2018), seafloor biomass (Wei et al., 2010), and seafloor total organic carbon (Wood et al., 2018; Lee et al., 2019) on a global scale.

In machine learning, a prediction model is constructed from a training dataset consisting of the target variable to be predicted, and a set of predictor variables. A random subset of the data, the test set, is typically held back for testing and validation of the prediction model. The most widely used machine learning methods are Artificial Neural Networks (e.g. Priddy and Keller,

2005), Support Vector Machines (Vapnik, 2000), and Random Forests (RF; Breiman, 2001).

RF is an ensemble classifier based on the concept of decision trees, which are grown from the training set by randomly drawing a subset of samples with replacement (bagging or bootstrap approach) (Breiman, 2001). At each tree node, the data are split based on a random subset of predictor variables to partition the data into relatively homogenous subsets and maximize the differences between the offspring branches. Each tree predicts on all samples in the test set and the final prediction is obtained

by averaging the predictions from all trees.

RF has been repeatedly found superior to other machine learning methods (e.g. Li et al., 2011; Cracknell and Reading, 2014). It is robust to noise and outliers (Breiman, 2001), and it is also able to handle high-dimensional and complex data. Moreover, RF does not require any preprocessing of the input variables and provides variable importance measurements, making it the first choice method in many applications.





Here, we apply RF to predict seismic p-wave velocity-depth profiles on a global scale, based on a set of 38 geological and spatial predictors that are freely available from global datasets. Prediction performance is evaluated and compared to velocity-depth profiles calculated from empirical $v_p$ functions. We also test additional methods for improvement of model performance and determine which predictors are most important for the prediction of $v_p$.

**2 Methods**

**2.1 Dataset**

**2.1.1 $v_p(z)$ data**

$v_p(z)$ profiles for training of the RF model were obtained from boreholes drilled by the DSDP, ODP and IODP campaigns between 1975 and 2016. All boreholes containing $v_p$ measurements were used, excluding those with bad-quality logs according

to the logging description notes. In total, 333 boreholes were included in the dataset, the distribution of which is shown in Fig. 1. All $v_p(z)$ data from these boreholes are available through http://www.iodp.org and were downloaded from the archive at http://mlp.ldeo.columbia.edu/logdb/scientific_ocean_drilling/.

A multitude of measuring methods and tools had been employed by the different drilling campaigns to obtain $v_p$ measurements, including wireline logging tools (e.g. sonic digital tool, long-spacing sonic tool, dipole sonic imager, borehole compensated

sonic tool) and logging-while-drilling tools (sonicVISION tool, ideal sonic-while-drilling tool). The majority of these methods provided $v_p$ measurements at 0.15 m depth intervals. Lengths of the $v_p$ logs varied greatly, ranging between 10 m and 1800 m (average: 370 m), with top depths of 0-1270 m (average: 138 m) and bottom depths of 16-2460 m (average: 508 m).

After exporting the $v_p(z)$ profiles for each borehole, the data were smoothed using a moving average filter with a window of 181 data points (corresponding to ca. 27 m for a 0.15 m depth interval). Smoothing was applied to remove outliers and to

20 account for unknown and varying degrees of uncertainty associated with the different measurement tools. In addition, smoothing was expected to facilitate prediction, as the aim was to predict the general $v_p(z)$ trend at a given location, rather than predicting exact $v_p$ values at a certain depth. Following smoothing, the profiles were sampled to 5 m depth intervals, using the same depth steps in all boreholes.

**2.1.2 Predictors**

A total of 38 geological and spatial variables obtained from the borehole metadata and freely available global datasets were included as predictors (Table 1). For predictor variables based on global grids, such as age of crust (crustage), sediment thickness (sedthick), and surface heatflow (heatflow), values were extracted for each borehole location in GMT (Wessel et al., 2013), using the command *grdtrack*. As the crustal age grid (Müller et al., 2008) contained only ages for oceanic crust, the age for locations above continental crust was set to 1 billion years to represent a significantly older age than that of oceanic crust.





Depth to basement (depth2base) was calculated by subtracting the depth values from the (constant) sedthick value at each borehole location, so that depths below the basement were indicated by a negative depth2base value. The distance predictor variables, e.g. distance to the nearest seamount (dist2smt), were calculated based on the borehole location and the respective datasets (Table 1) via the GMT command *mapproject*.

Of the 38 predictors, 15 were of the type continuous, whereas 23 were categorical variables describing the type of crust and the geological setting at each borehole location (Table 1). The categorical predictors were encoded as either 0 or 1, depending on whether the predictor corresponded to the geological setting at a given borehole. Multiple categories were possible; for example, a borehole located in a fore-arc basin above continental crust would be described by 1 for the predictors "contcrust", "active_margin", "subduction" and "fore-arc", and 0 for all other categorical predictors.

**2.2 Random Forest implementation**

RF was implemented using the *RandomForestRegressor* in Python's machine learning library scikit-learn (Pedregosa et al., 2011). Two parameters needed to be set: the number of trees (n_estimators) and the number of randomly selected predictors to consider for splitting the data at each node (max_features). Many studies used 500 trees (e.g. Micheletti et al., 2014; Belgiu and Drăguţ , 2016; Meyer et al., 2017, 2018), but as performance still increased after 500 trees, we chose 1000 trees instead.

The max_features parameter was initially set to all predictors (38), as recommended for regression cases (Pedregosa et al., 2011; Müller and Guido, 2017), although some studies suggest tuning this parameter to optimize model results (Micheletti et al., 2014; Ließ et al., 2016; Meyer et al., 2016b).

**2.3 Model validation**

A 10-fold cross-validation (CV), an approach frequently used in model validation (e.g. Li et al., 2011; Gasch et al., 2015; Ließ
et al., 2016; Meyer et al., 2016b, 2018), was applied to validate the RF model. CV involved partitioning the dataset into ten equally sized folds. Nine of these folds acted as the training set used for model building, whereas the remaining fold was used for testing the model and evaluating the performance. This procedure was repeated so that each fold acted once as the test fold, and hence each borehole was once part of the test set. Performances of all test folds were averaged to give a final model performance.

Partitioning into folds was not done randomly from all available data points but by applying a leave-location-out (LLO) approach (Gasch et al., 2015; Meyer et al., 2016a, 2018) in which the data remained separated into boreholes, i.e., locations, so that each fold contained 1/10 of the boreholes. With 33-34 boreholes per fold, the size of the training dataset thus varied between 20166 and 20784 data points. By using the LLO approach, whole locations were left out of the training set, thereby allowing the RF model to be tested on unknown locations through prediction of $v_p$ for each borehole in the test fold. If the
folds were chosen randomly from all data points, each borehole location would be represented in the training set by at least some data points, resulting in overoptimistic model performance due to spatial overfitting (Gasch et al., 2015; Meyer et al., 2016a, 2018).



Performance of the RF model was evaluated by comparing the predicted and true $v_p(z)$ curves for each borehole in the test fold and calculating the standard error metrics root mean square error (RMSE), mean absolute error (MAE), and the coefficient of determination ($R^2$). RMSE, MAE and $R^2$ were then averaged over the ten folds.

In addition, we also tested how well the predicted $v_p(z)$ curves performed compared to $v_p(z)$ curves calculated from empirical

functions. Using the depth values of the respective test borehole, $v_p(z)$ profiles were therefore calculated from the five empirical functions presented by Hamilton (1985) for deep-sea sediments, i.e., for terrigenous silt and clays (termed H1 in the following), terrigenous sediments (H2), siliceous sediments (H3), calcareous sediments (H4), and pelagic clay (H5). These $v_p$ functions were chosen because the deep-sea setting applied to the majority of the boreholes, or was the best choice in absence of empirical functions for other geological settings such as mid-ocean ridges. The resulting Hamilton curves were evaluated against the

true $v_p(z)$ profile, and RMSE, MAE and $R^2$ were averaged over the five curves. The averaged error metrics were then compared to the error metrics of the prediction, and each borehole was assigned a score between 0 and 3 as shown in Table 2. Scores 2 and 3 were interpreted as a good prediction, i.e., better than the Hamilton curves, whereas scores 0 and 1 represented generally bad predictions. The proportion of boreholes with good predictions, averaged over the ten folds, served as another performance evaluation measure.

**2.4 Predictor selection**

To determine the most important predictors for $v_p$ prediction, a predictor selection approach was performed. Although RF can deal with high data dimensionality, predictor selection is still recommended, not only to remove predictors that could cause overfitting but also to increase model performance (e.g. Belgiu and Drăguţ, 2016, and references therein). We applied Recursive Feature Elimination (RFE), which is based on the variable importance scores provided by the RF algorithm. After

calculating and evaluating a model with all 38 predictors, the least important predictor according to the variable importance scores was removed and the model was calculated again. This procedure was repeated until only one predictor was left. By evaluating model performance for each run via CV, using the same ten folds as before, the optimum number of predictors was determined.

**2.5 Tests to improve prediction performance**

Additional tests to improve prediction performance included predictor scaling, variation of the max_features parameter, and stronger smoothing of the $v_p(z)$ curves. All models were evaluated via a 10-fold CV, using the same folds as in the previous model runs.

Predictor scaling was applied to account for the different data ranges of continuous and categorical features. Model performance may be negatively affected if different types of variables or data ranges are used (Otey et al., 2006; Strobl et al.,

2007), even though RF does not normally require scaled input data. All continuous predictors were scaled to between 0 and 1 to match the range of the categorical predictors, and RFE was repeated.

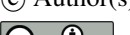


As tuning of the max_features parameter, i.e., the number of predictors to consider at each split, is recommended by some studies (Ließ et al., 2016; Meyer et al., 2018), an additional model was run in which max_features was varied between 2 and 38 (all features) with an interval of 2. Performance was evaluated for each case to find the optimum number of predictors to choose from at each split.

A third attempt to improve model performance involved enhanced filtering of the $v_p(z)$ curves so that larger $v_p$ variations were smoothed out and the curves indicated only a general trend, which would likely be sufficient for many applications requiring knowledge of $v_p$ with depth. The $v_p$ curves therefore underwent spline smoothing using Python's scipy function *UnivariateSpline*. Three separate RF models were calculated: (i) spline1, which involved spline smoothing of the predicted curve of each test borehole; (ii) spline2, in which the input $v_p(z)$ data were smoothed; and (iii) spline3, where both the input

$v_p(z)$ curves and the predictions were smoothed. All three cases were run with the 16 most important predictors as determined from the RFE results, and compared to the previous models.

## 3 Results

### 3.1 Prediction performance

Overall, many $v_p(z)$ profiles were predicted well by the RF models. For the 38-predictor CV, about 59.5 % of the boreholes

had prediction scores of 2 or 3, representing a prediction performance superior to that of the Hamilton functions.

Predictions of prediction score 3, which were characterised by lower RMSE and MAE values and a higher $R^2$ than the five empirical functions, often exhibited a good fit to the true $v_p(z)$ curve (Fig. 2a-d). Even for more complex velocity profiles, e.g. involving a velocity reduction at depth (Fig. 2d), or a strong increase such as that from 2.2 km s$^{-1}$ to >4 km s$^{-1}$ at the basement contact in Fig. 2b, the predicted $v_p(z)$ curves generally matched the true curves well. In some cases, score 3 predictions did not

provide a good fit but still performed better than the empirical functions (Fig. 2e). Score 2 predictions generally indicated the correct trend of the true $v_p(z)$ profile (Fig. 2f), whereas score 1 and score 0 predictions failed to do so, with velocities often considerably higher or lower than the true velocities (Fig. 2g, h).

The RFE CV revealed best performance for 33 predictors, as indicated by the lowest RMSE and MAE values (Fig. 3a). The proportion of boreholes with prediction scores of 2 or 3 was 59.2 % and thus slightly lower than for the 38-predictor CV (59.5

25   %; Fig. 3b). The highest proportion of 61.9 % was achieved by the 16-predictor model (Fig. 3b), but this also led to the highest errors (Fig. 3a).

By scaling all predictors to between 0 and 1 and repeating RFE, RMSE and MAE were reduced further, with the best errors obtained for 35 predictors (Fig. 3a). These errors were only slightly lower than those of the 30-predictor case, which achieved a higher percentage of boreholes with good prediction (60.4 %; Fig. 3b).

Varying the number of predictors to consider for splitting the data at each tree node also improved the performance. For max_features = 22, RMSE and MAE were lower than in all previous RF cases (Fig. 3a), while the proportion of boreholes



with good performance was 61.3 % and thus only slightly lower than for the 16-predictor case in which all 38 predictors were considered (Fig. 3b).

The three attempts of stronger smoothing of the $v_p(z)$ profiles via splines resulted in overall worse performance than the 16-predictor case, both in terms of errors and proportion of well-predicted boreholes (Fig. 4a). An exception is the spline1 case

(spline smoothing of the predicted $v_p(z)$ profile), for which 62.4 % of the boreholes had scores of 2 or 3 (Fig. 4b), although RMSE and MAE were slightly worse than for the other RF cases.

### 3.2 Score distribution

The global distribution of boreholes with different prediction scores, shown in Fig. 5 for the 16-predictor case without spline smoothing, did not indicate a clear separation into areas with relatively good (scores 2 and 3) or bad (scores 0 and 1) prediction

performance. Some areas contain clusters of >10 boreholes, many of which had a prediction score of 3. Examples included the Sea of Japan (area A in Fig. 5a), the Nankai Trough (B), the Ontong-Java Plateau (C), the Queensland Plateau (D), and the Great Australian Bight (E). However, nearly all of these cluster areas also contained boreholes with bad prediction scores (Fig. 5b). Similarly, single boreholes in remote locations were often characterised by a prediction score of 0 (Fig. 5b), but there were also several remote boreholes with scores of 3, e.g. on the Mid-Atlantic Ridge (area F in Fig. 5a).

### 3.3 Predictor importance

For the 38-predictor CV, the five most important predictors were "depth2base", "crustage", "depth", "dist2smt", and "wdepth" (Fig. 6). Continuous predictors and categorical predictors were clearly separated in the predictor importance plot (Fig. 6), with continuous predictors being of high importance in the RF model, whereas categorical predictors appeared less important. The only exception was the categorical predictor variable "spreading_ridge", which had a slightly higher importance ranking than

the continuous predictors "long" and "dist2transform". Many of the categorical predictors were of negligible (almost 0) importance (Fig. 6).

When the least important predictor was eliminated after each model run using RFE, the same trend was observed: in both the unscaled and scaled RFE cases, all categorical predictors were eliminated before the continuous predictors (Table 3). In the 16-predictor case, which had the highest proportion of well-predicted boreholes (61.9 %), the only categorical predictor

included was "spreading_ridge".

In the unscaled RFE case, the five most important predictors were the same as in the feature importance plot of the 38-predictor case (Fig. 6). However, the order differed slightly, with "depth" being eliminated before "dist2smt", "wdepth", "depth2base", and "crustage" (Table 3). When using scaled predictors, the five top predictors included "heatflow" (ranked sixth in both the 38-predictor CV and unscaled RFE cases) instead of "crustage". "Crustage" dropped to position 15 and was thus the least

important of the continuous predictors (Table 3). In general, however, the position ranking of most predictors varied only by up to five positions between the unscaled and the scaled RFE cases (Table 3).





## 4 Discussion

### 4.1 Prediction performance in comparison with empirical functions

Our results show that $v_p(z)$ profiles can be predicted successfully using machine learning. Overall, the applied RF approach is superior to the empirical $v_p$ functions of Hamilton (1985), as indicated by the 60 % of tested boreholes with prediction scores

of 3 or 2. Although such a quantitatively better performance (i.e., lower RMSE and MAE, and higher $R^2$ than the Hamilton $v_p(z)$ profiles) does not always mean a perfect fit to the true $v_p(z)$ curve of the tested borehole, the RF approach has a promising potential for the prediction of $v_p$ with depth.

Slight improvements of the prediction performance were achieved by applying RFE, resulting in a proportion of well-predicted boreholes of 61.9 % for the 16-predictor model. Smoothing the predicted $v_p(z)$ profiles via spline smoothing (spline1 case)

provided a further increase to 62.4 % of well-predicted boreholes. In addition, reducing the max_features parameter from 38 (all predictors) to 22 also resulted in a slight improvement (61.3 %), thus supporting other studies that recommended tuning the max_features parameter to improve results (Ließ et al., 2016; Meyer et al., 2018). However, to increase model performance even further, to a proportion of well-predicted boreholes well exceeding 60 %, other changes are required.

### 4.2 Most important predictors for the prediction of $v_p(z)$

Both the predictor importance ranking of RF and the RFE results revealed "depth" as one of the most important predictors. However, "depth" was not the most important predictor, which is surprising as empirical $v_p$ functions, including those of Hamilton (1985), all use depth as the only input parameter. Our results showed that "depth2base" was always ranked higher than "depth", and often the predictors "wdepth", "dist2smt" and "crustage" also had higher importance scores than "depth". Although "depth" is obviously still an important parameter in the prediction of $v_p$, these observations imply that empirical

functions using only depth as input and neglecting all other influences may not produce realistic $v_p$ values, which is supported by the at least 60 % of test locations for which the RF approach produced better $v_p(z)$ profiles than the Hamilton functions.

The high importance of the predictors "depth2base", "wdepth", "dist2smt", "crustage", as well as "heatflow", seems reasonable. The depth to the basement, which is related to the sediment thickness, is relevant because of the rapid $v_p$ increase at the basement contact and the associated transition from relatively low (<2.5 km s$^{-1}$) to higher (> 4 km s$^{-1}$) $v_p$ values. Even

though in the majority of boreholes, the basement was not reached, the depth to the basement strongly influences $v_p$. The high ranking of the distance to the neareast seamount is likely attributed to the associated change in heatflow at seamount locations. Higher heatflow and hence higher temperatures affect density, which in turn affects $v_p$. The predictor "crustage" indicates young oceanic crust, which is characterised by higher temperature and hence lower density, affecting $v_p$. Moreover, "crustage" differentiates between oceanic (<200 Myr) and continental (here: 1 Byr) crust, and apparently more effectively than the

categorical predictors "oceancrust" and "contcrust", which are of considerably lower importance.



It has to be noted that the high-importance predictors discussed above only represent the most important of the 38 predictors used for prediction of $v_p$; they are not necessarily the parameters that most strongly influence $v_p$ in general. If other parameters, such as porosity, density, pressure, or saturation, had been included as predictors, they would likely have resulted in a higher importance ranking than, e.g., "dist2smt" or "crustage". However, these parameters were not included in the model as they

were restricted to measurements at borehole locations – not necessarily those from which $v_p(z)$ data were obtained – and are therefore not available for every location in the oceans. For the same reason, other geophysical parameters, e.g. electrical resistivity and magnetic susceptibility, were also not included.

A surprising finding in terms of predictor importance is the low importance of all categorical predictors. The clear separation between continuous and categorical predictors in the predictor importance plots may be due to biased predictor selection, as

observed by Strobl et al. (2007) when different types of predictors were used. In such cases, categorical predictors may often be neglected and ignored by the machine learning algorithm (Otey et al., 2006). Scaling the continuous predictors to the same range as the categorical predictors did not help to change the importance ranking, but bias cannot be excluded. On the other hand, it is also possible that the geological setting described by the categorical predictors was simply not relevant to the prediction of $v_p$. This possibility appears to be supported by the RFE results, which reveal the best performance (61.9 % of

well-predicted boreholes) when all but one categorical predictors were excluded (16-predictor case).

### 4.3 Suggestions for further improvement of performance

The fact that prediction performance could not be much improved by predictor selection, tuning the max_features parameter, or additional smoothing suggests that other measures are needed to further improve the prediction performance. The comparatively high proportion of boreholes with badly predicted $v_p(z)$ profiles (about 40 %) is likely due to the limited number

of boreholes that were available in this study, but may also have been influenced by the choice of machine learning algorithm. It is possible to add more predictors that potentially influence $v_p$, for example, seafloor gradient, bottom water temperature, and distance to the shelf edge. In addition, some of the predictors could be improved. For example, the age of the continental crust, currently set to the constant value of 1 Byr, could be adapted based on the crustal age grid by Poupinet and Shapiro (2009). Other studies also suggest including the first and second derivatives of predictors or other mathematical combinations

of predictors (Obelcz and Wood, 2018; Wood et al., 2018; Lee et al., 2019).

Another way to extend the dataset is to include more $v_p(z)$ data. Given the relatively inhomogenous global distribution of borehole locations used in this study (Fig. 1), adding more $v_p(z)$ data is highly recommended. On a much smaller scale, Gasch et al. (2015) noted that high spatial heterogeneity of input locations limits the prediction performance and increases prediction errors. Adding more $v_p(z)$ data, especially from regions such as the southern Pacific and Atlantic oceans that are presently not

covered, will likely help to increase the prediction performance. For example, the $v_p(z)$ records from recent IODP expeditions may be added to the dataset as they become available. Additional $v_p$ data could also be obtained from commercial boreholes and refraction seismic data from ocean bottom seismometers, although the latter would be of lower vertical resolution.





The choice of machine learning algorithm may also influence model performance. Studies comparing RF against other machine learning algorithms reported different trends: in some cases, RF was superior in terms of prediction performance (e.g. Li et al., 2011; Cracknell and Reading, 2014), whereas in other cases, no strong differences were observed between the different methods (e.g. Goetz et al., 2015; Meyer et al., 2016b). Given the generally positive reputation of RF as a prediction method,

we doubt that a different algorithm would lead to a significantly different prediction performance for $v_p$.

## 5 Conclusions

In this study, we presented an RF model for the prediction of $v_p(z)$ anywhere in the oceans. In about 60 % of the tested locations, the RF approach produced better $v_p(z)$ profiles than empirical $v_p$ functions. This indicates a promising potential for the prediction of $v_p(z)$ using machine learning, although some improvement is still required. In particular, the model input data

should be extended to increase spatial coverage, which is expected to significantly improve prediction performance. Our results showed that depth, which is the only input in empirical $v_p$ functions, is not the most important parameter for the prediction of $v_p$. Distance to the basement, water depth, age of crust, and distance to the nearest seamount are, in general, equally or even more important than depth. By including these parameters in the determination of $v_p$, the RF model is able to produce more accurate $v_p(z)$ profiles and can therefore be used as an alternative to empirical $v_p$ functions. This is of particular interest for

geophysical modelling applications, such as modelling gas hydrate concentrations, in areas lacking alternative $v_p(z)$ information from boreholes or seismic data.

## Acknowledgements

This study was funded by the Helmholtz Association, grant ExNet-0021.

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



**Author contributions:** ID performed the modelling and analyses and CB acquired funding for the project. ID prepared the manuscript with contributions from CB.

**Competing interests:** The authors declare that they have no conflict of interest.

**Figure captions**

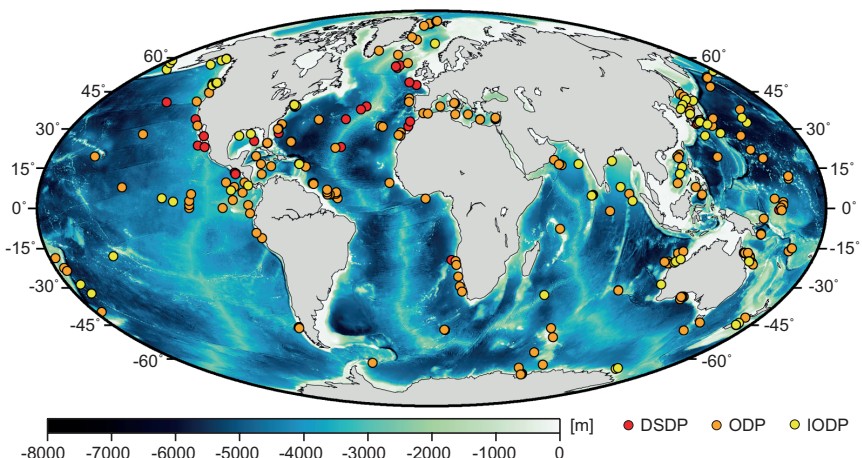

**Figure 1. Distribution of the 333 boreholes from which $v_p(z)$ profiles were extracted. DSDP – Deep Sea Drilling Project, ODP –**

10  **Ocean Drilling Program, IODP – International Ocean Discovery Program. Bathymetry (30 s resolution) is from the GEBCO_2014 grid (http://www.gebco.net).**



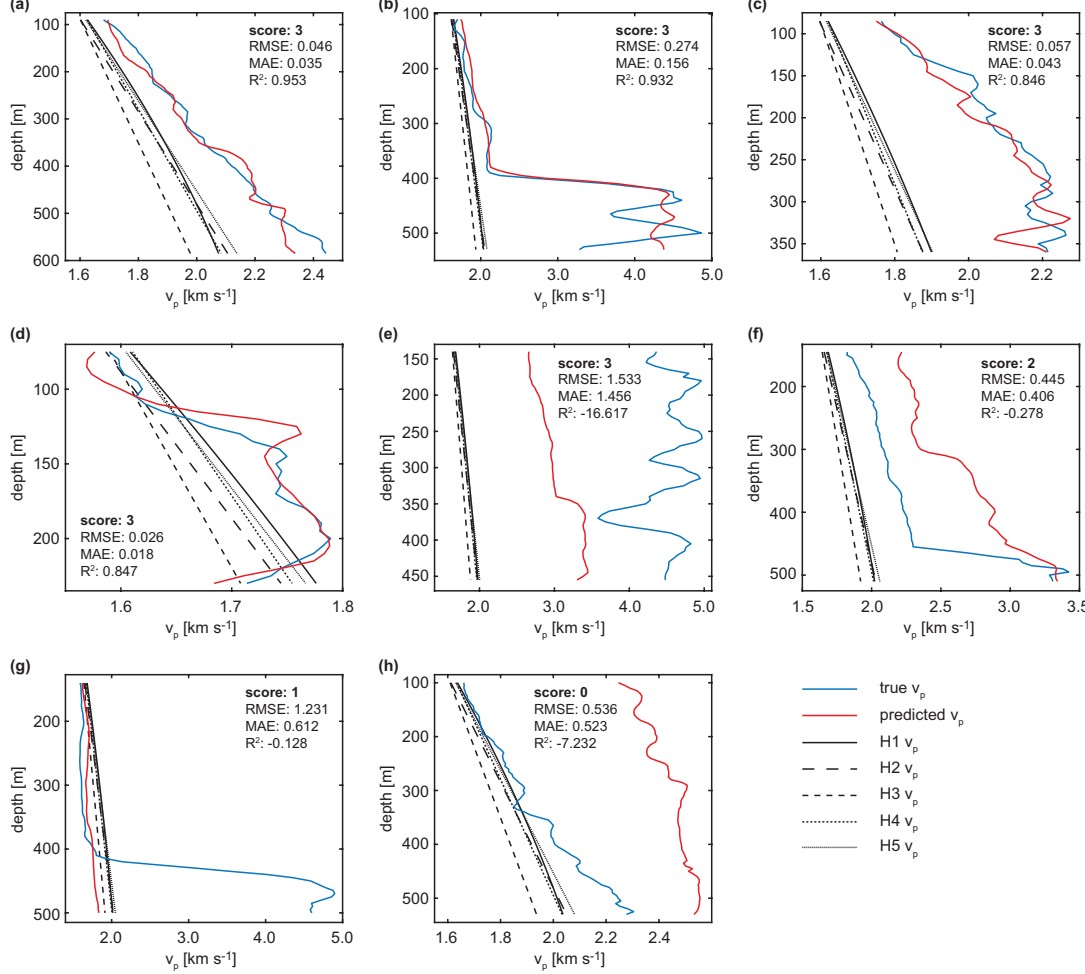

**Figure 2.** Examples for true $v_p(z)$ curves, predicted $v_p(z)$ curves, and $v_p(z)$ calculated from the five Hamilton functions (Hamilton, 1985) used in model evaluation. (a)-(d) well predicted $v_p(z)$ curves of score 3, (e) less good prediction of score 3, (f) score 2, (g)-(h) bad predictions of scores 1 and 0. See 2.3 for a description of H1 to H5.




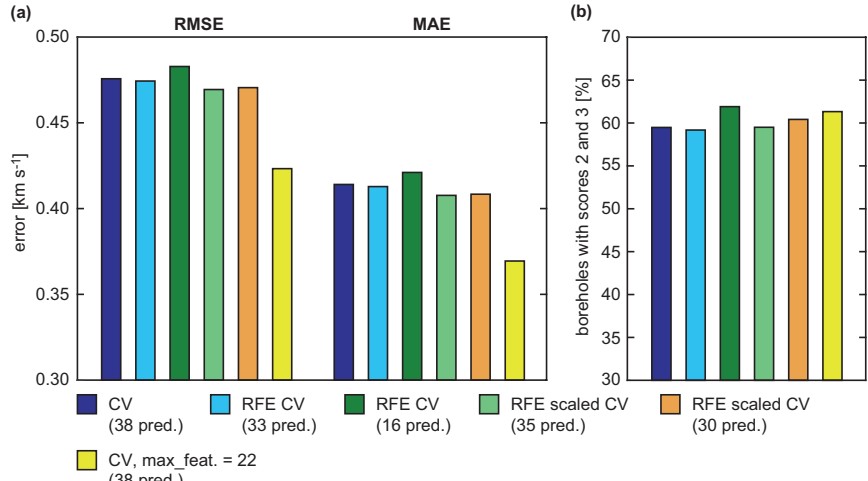

**Figure 3.** Comparison of (a) error metrics and (b) proportion of well predicted boreholes (scores 2 and 3) for different model runs.
RMSE – root mean square error, MAE – mean absolute error, CV – cross-validation, RFE – Recursive Feature Elimination.

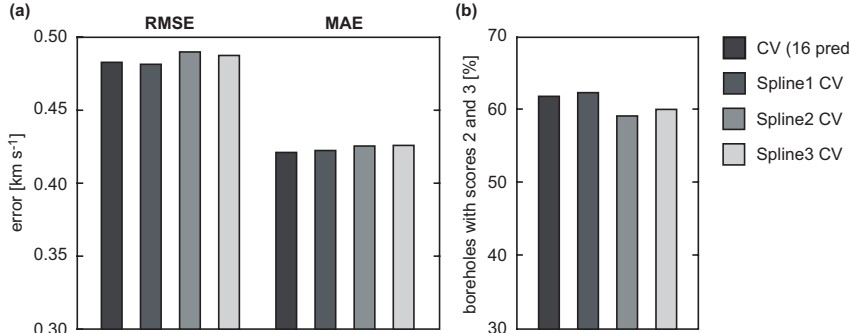

**Figure 4.** Comparison of (a) error metrics and (b) proportion of well predicted boreholes (scores 2 and 3) for model runs with
different degrees of data smoothing. RMSE – root mean square error, MAE – mean absolute error, CV – cross-validation.



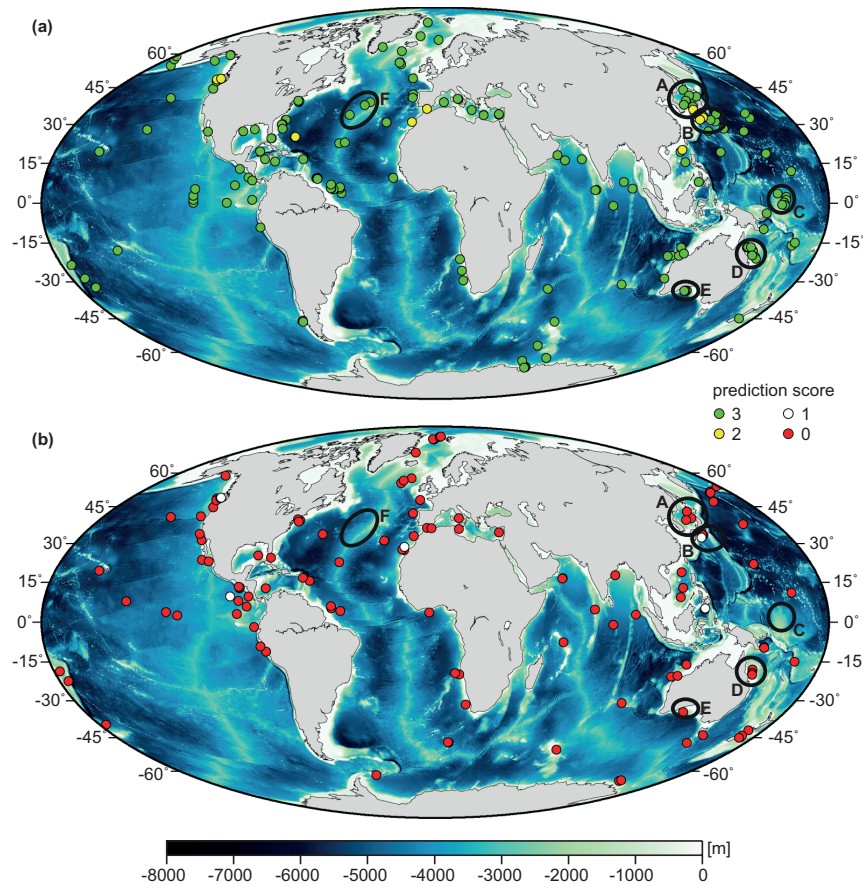

**Figure 5. Distribution of boreholes with (a) good (scores 2 and 3) and (b) bad (scores 0 and 1) $v_p$ predictions. Areas A-E mark clusters of boreholes in the Sea of Japan (A), the Nankai Trough (B), the Ontong-Java Plateau (C), the Queensland Plateau (D), and in the Great Australian Bight (E). Area F indicates an example for remote boreholes of score 3 on the Mid-Atlantic Ridge. Bathymetry (30 s resolution) is from the GEBCO_2014 grid (http://www.gebco.net).**





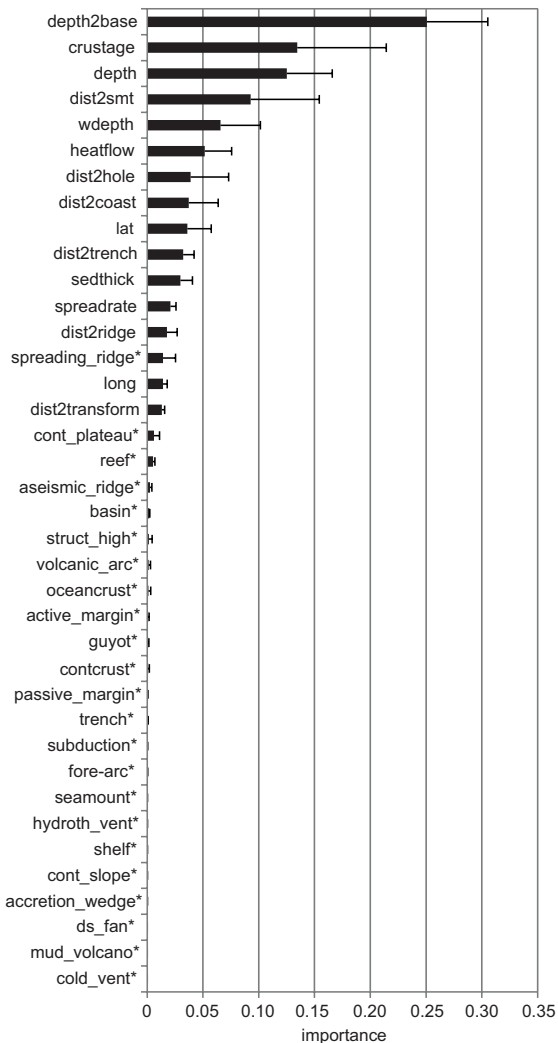

**Figure 6. Predictor importance ranking for the 38-predictor model run. For each predictor, the importance was averaged over the ten runs of the 10-fold CV. Categorical predictors are marked with an asterisk. Predictor names are explained in Table 1.**



**Table 1.** Overview of the 38 predictors and their sources.

| predictor | description | type | source description | reference |
|---|---|---|---|---|
| lat | latitude | continuous | DSDP/ODP/IODP data processing notes | |
| long | longitude | continuous | DSDP/ODP/IODP data processing notes | |
| wdepth | water depth | continuous | DSDP/ODP/IODP data processing notes | |
| depth | depth below seafloor | continuous | $v_p$ logs | |
| crustage | age of crust | continuous | ocean crust: global crustal age grid (2 min res.) | Müller et al. (2008) |
| | | | continental crust: 1 Byr (const.) | |
| sedthick | sediment thickness | continuous | global sediment thickness grid (5 min res.) | Whittaker et al. (2013) |
| spreadrate | spreading rate | continuous | global spreading rate grid (2 min res.) | Müller et al. (2008) |
| heatflow | surface heatflow | continuous | global surface heatflow grid (2° res.) | Davies (2013) |
| depth2base | depth to acoustic basement | continuous | derived from sediment thickness and depth | |
| dist2smt | distance to nearest seamount | continuous | derived from global seamount dataset | Kim and Wessel (2011) |
| dist2hole | distance to nearest borehole | continuous | derived from borehole locations | |
| dist2coast | distance to nearest coast | continuous | derived from global shoreline dataset | Wessel and Smith (1996) |
| dist2trench | distance to nearest trench | continuous | derived from global trench dataset | Coffin et al. (1998) |
| dist2ridge | distance to nearest spreading ridge | continuous | derived from global spreading ridge dataset | Coffin et al. (1998) |
| dist2transform | distance to nearest transform boundary | continuous | derived from global transform boundary dataset | Coffin et al. (1998) |
| oceancrust | oceanic crust | categorical | derived from crustal age | |
| contcrust | continental crust | categorical | derived from crustal age | |
| active_margin | geological setting: active margin | categorical | DSDP/ODP/IODP proceedings (site descriptions) | |
| passive_margin | geological setting: passive margin | categorical | DSDP/ODP/IODP proceedings (site descriptions) | |
| spreading_ridge | geological setting: spreading ridge | categorical | DSDP/ODP/IODP proceedings (site descriptions) | |
| subduction | geological setting: subduction zone | categorical | DSDP/ODP/IODP proceedings (site descriptions) | |
| volcanic_arc | geological setting: volcanic arc | categorical | DSDP/ODP/IODP proceedings (site descriptions) | |
| fore-arc | geological setting: fore-arc basin | categorical | DSDP/ODP/IODP proceedings (site descriptions) | |
| accretion_wedge | geological setting: accretionary wedge | categorical | DSDP/ODP/IODP proceedings (site descriptions) | |

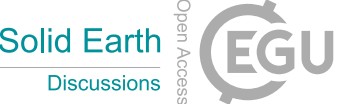

| trench | geological setting: trench | categorical | DSDP/ODP/IODP proceedings (site descriptions) | |
|---|---|---|---|---|
| cont_slope | geological setting: continental slope | categorical | DSDP/ODP/IODP proceedings (site descriptions) | |
| shelf | geological setting: continental shelf | categorical | DSDP/ODP/IODP proceedings (site descriptions) | |
| reef | geological setting: (former) reef | categorical | DSDP/ODP/IODP proceedings (site descriptions) | |
| basin | geological setting: basin | categorical | DSDP/ODP/IODP proceedings (site descriptions) | |
| struct_high | geological setting: structural high | categorical | DSDP/ODP/IODP proceedings (site descriptions) | |
| cont_plateau | geological setting: continental plateau | categorical | DSDP/ODP/IODP proceedings (site descriptions) | |
| aseismic_ridge | geological setting: aseismic ridge | categorical | DSDP/ODP/IODP proceedings (site descriptions) | |
| seamount | geological setting: seamount | categorical | DSDP/ODP/IODP proceedings (site descriptions) | |
| guyot | geological setting: guyot | categorical | DSDP/ODP/IODP proceedings (site descriptions) | |
| mud_volcano | geological setting: mud volcano | categorical | DSDP/ODP/IODP proceedings (site descriptions) | |
| ds_fan | geological setting: deep-sea fan | categorical | DSDP/ODP/IODP proceedings (site descriptions) | |
| hydroth_vent | geological setting: hydrothermal vent | categorical | DSDP/ODP/IODP proceedings (site descriptions) | |
| cold_vent | geological setting: cold vent | categorical | DSDP/ODP/IODP proceedings (site descriptions) | |



**Table 2. Scores for performance comparison between RF prediction and v$_p$ calculated from the empirical functions of Hamilton**
5 **(1985).**

| Score | Description | Inferred prediction performance |
|---|---|---|
| 3 | all 3 error metrics of RF prediction indicate better fit than empirical functions | Good |
| 2 | 2 of 3 error metrics of RF prediction indicate better fit than empirical functions | Good |
| 1 | 1 of 3 error metrics of RF prediction indicate better fit than empirical functions | Bad |
| 0 | all 3 error metrics of empirical functions indicate better fit than RF prediction | Bad |




**Table 3. Predictor ranking based on the RFE results for unscaled and scaled predictors. Categorical predictors are marked with an asterisk. See Table 1 for an explanation of predictor names.**

| Position | Predictor | |
|---|---|---|
| | RFE unscaled | RFE scaled |
| 1 | Crustage | wdepth |
| 2 | depth2base | depth2base |
| 3 | Wdepth | dist2smt |
| 4 | dist2smt | depth |
| 5 | Depth | heatflow |
| 6 | Heatflow | sedthick |
| 7 | dist2hole | dist2trench |
| 8 | dist2coast | dist2hole |
| 9 | dist2trench | dist2coast |
| 10 | Lat | spreadrate |
| 11 | Sedthick | dist2ridge |
| 12 | Spreadrate | long |
| 13 | dist2ridge | lat |
| 14 | Long | dist2transform |
| 15 | dist2transform | crustage |
| 16 | spreading_ridge* | contcrust* |
| 17 | cont_plateau* | basin* |
| 18 | reef* | active_margin* |
| 19 | aseismic_ridge* | struct_high* |
| 20 | basin* | oceancrust* |
| 21 | struct_high* | passive_margin* |
| 22 | oceancrust* | subduction* |
| 23 | volcanic_arc* | reef* |
| 24 | active_margin* | accretion_wedge* |
| 25 | contcrust* | cont_plateau* |
| 26 | guyot* | cont_slope* |
| 27 | passive_margin* | spreading_ridge* |
| 28 | trench* | fore-arc* |
| 29 | subduction* | shelf* |
| 30 | seamount* | ds_fan* |
| 31 | fore-arc* | volcanic_arc* |
| 32 | hydroth_vent* | trench* |
| 33 | cont_slope* | seamount* |
| 34 | shelf* | aseismic_ridge* |
| 35 | accretion_wedge* | guyot* |



| 36 | ds_fan* | cold_vent* |
|----|---------|------------|
| 37 | mud_volcano* | hydroth_vent* |
| 38 | cold_vent* | mud_volcano* |

