# Peer review of "Prediction of seismic p-wave velocity using machine learning"

_Solid Earth, 2019_

## Referee Comment (RC1) · Taylor Lee (Referee) · 7 May 2019

**Prediction of seismic p-wave velocity using machine learning**
**Ines Dumke and Christian Berndt**
**Solid Earth**
**Referee Comment - Taylor Lee**
* * *
**General comments**

Machine learning has been previously well established in other fields, but has not grasped attention in a similar way within the geosciences. This paper uses sparse p-wave velocity data from DSDP/ODP/IODP as training data in a machine learning algorithm (Random Forest) to predict p-wave velocity with depth. A thorough analysis was done to determine how effective machine learning is at predicting vertical velocity profiles. This analysis included comparison of p-wave velocity machine learning predictions with empirical estimates. A variety of appropriate methods were tested to improve the machine learning prediction (e.g. smoothing input data and prediction results, varying max_features and number of predictors used, 10-fold cross validation, predictor value scaling). As a result, this work provides valuable information on types of useful predictors and variables highly correlated to p-wave velocity. Additionally, this method shows in some case superior to using strictly empirical methods to estimate p-wave velocity with depth.

Results show this work is novel and useful. However, there is a major component of the analysis missing. This work contains many examples of validation of previously existing p-wave velocity but lacks demonstration on prediction of p-wave velocity in areas where no velocity data is available.

**Specific comments**

Page 3 Section 2.1.2 (Predictors) Line 28 mentions that the continental crust was set at 1 billion years to represent significant older crust than that of the oceanic crust. If all the observed data (DSDP/ODP/IODP) are on oceanic crust, what is the importance/meaning of defining continental crust age?

Page 7 Section 3.3 (Predictor importance) Line 20 states that categorical predictors generally do not have any importance in prediction performance. Additionally, it is again discussed in the discussion section (Section 4.2- lines 8-14 page 9). What is the variance of your sampled data set in categorical predictors? For example, for a given test data set (i.e. fold) are all of your categorical predictors for that run a 1 or 0? If all of your test data set has only one categorical value then that predictor would be of no importance.

Consider, if true, explicitly stating that predictions of this kind have not done with depth before. (page 2 ~ lines 16-20)

Minor suggestion to add in the abstract that this method is not designed to capture high variance in a p-wave velocity profile, but is instead intended to capture the overall trend of p-wave velocity profile.

It is stated and supported (Line 1 page 7; Figure 3) that the RFE CV 16 predictors prediction (green) is better than CV, max_features =22, 38 predictors however the error in the prediction is significantly higher for the green prediction with roughly the same % boreholes labelled as "good". Why do you consider green prediction to be so much better than yellow prediction? It might be useful if you explicitly state what your ultimate metric of correctness is (e.g. highest % correct or lowest error?)

What is the final global spatial resolution? E.g. prediction of p-wave velocity profile every 1-degree, 5-min, etc.?

Page 9 Section 4.2 (Most important predictors for the prediction of $v_p(z)$) Lines 2-8 discuss how certain predictors are not used (porosity, density, pressure) as not all boreholes have depth associated measurements. However, some of the predictors used in the prediction do not have a depth component (e.g. crustage). Applying this logic, why do you not use seafloor porosity (i.e. depositional porosity) or likewise predictors?

No supplemental material was provided for the global prediction of p-wave velocity with depth. This paper should include the final global prediction of p-wave velocity with depth.

**Technical corrections**

Page 8 delete "the" on line 21: "by the at least 60% of test locations"

Page 8 line 3 consider changing "our results show that $v_p(z)$ profiles" to "our results show that the general trend of vp(z) profiles"

Page 16 Figure 2 caption (e) change "less good" to different word (substandard?)

Page 23 table 3, change words so they have consistent capitalization between table columns (e.g. Long and long)

Page 12 Lee et al., 2019 citation is missing the publication year.

---

## Referee Comment (RC2) · Anonymous Referee #2 · 15 Jul 2019

The manuscript, " Prediction of seismic p-wave velocity using machine learning", is a well-written description of a machine learning method –Random Forests – to predict seismic p-wave velocity as a function of depth for any a generic marine location. This manuscript is suitable for Copernicus, but the manuscript needs to be revised before it can be accepted. I have some suggestions here.

1 Introduction.

Page 2:

L24: You make the statement that the most widely used machine learning methods are ANNs, SVMs, and RFs. It is hard to convince people that these three algorithms are the most widely used. For specific problems, some algorithms may be more common

than the other algorithms. You may say that the most widely used machine learning includes ANNs, SVMs, and RFs.

L31: You mentioned that RF has been repeatedly found superior to other machine learning methods. You need to specify the particular problems that RF has been found superior to "other machine learning methods" in the text. And what other machine learning methods do you mean here? Please specify in the text.

2 Methods

Page 3, section 2.1.2:

L25: How do you come up with these 38 predictors? Could you specify the reason why you choose these 38 predictors in this section?

Page 4, section 2.2:

L14: How do you define "performance"? I saw you mentioned performance in the later section 2.3. But it is better to define that when you first mention that. In addition, why do you choose 1000 trees? what is the maximum depth of each tree? How does the number of trees and depth affect the bias and variance of the prediction?

3 Results

Page 6, section 3.1:

The performance of an algorithm should be shown by both bias and variance. I only see the comparison of errors and percentage of boreholes with scores 2 and 3 in Fig. 3 and 4. How does the number of predictors and data smoothing affect the variance of the prediction?

Since you only have 333 boreholes, 2% change due to different model runs only change scores of $\sim$7 boreholes. I am curious about the location distributions of those boreholes which changed their scores, and why their scores changed by changing the number of predictors or data smoothing.

4 Discussion

Page 10:

L1-5: You made a strong statement about performance of RF. As I suggested in your introduction section, the performance of a machine learning algorithm really depends on situations.

5 Conclusion

Page 10:

L15: RF is hard to extrapolate to data outside the range they have been seen. I doubted that RF can be used for geophysical modeling in areas lacking v_p(z) from boreholes or seismic data.

---

## Author Comment (AC1) · 9 Aug 2019

General comments

Machine learning has been previously well established in other fields, but has not grasped attention in a similar way within the geosciences. This paper uses sparse p-wave velocity data from DSDP/ODP/IODP as training data in a machine learning algorithm (Random Forest) to predict p-wave velocity with depth. A thorough analysis was done to determine how effective machine learning is at predicting vertical velocity profiles. This analysis included comparison of p-wave velocity machine learning predictions with empirical estimates. A variety of appropriate methods were tested to improve the machine learning prediction (e.g. smoothing input data and prediction re-

sults, varying max_features and number of predictors used, 10-fold cross validation, predictor value scaling). As a result, this work provides valuable information on types of useful predictors and variables highly correlated to p-wave velocity. Additionally, this method shows in some case superior to using strictly empirical methods to estimate p-wave velocity with depth. Results show this work is novel and useful. However, there is a major component of the analysis missing. This work contains many examples of validation of previously existing p-wave velocity but lacks demonstration on prediction of p-wave velocity in areas where no velocity data is available.

AC: As we explain in the Methods section, due to our leave-location-out approach all predictions are made for locations that were withheld from the training data and therefore act as unknown locations. Validation of the prediction involved a comparison against the true vp data, but these data were in no case part of the prediction model. We explain below (last "specific comment") why we refrain from making predictions for completely new locations as it is beyond the scope of this paper, i.e. the purpose of this paper is to demonstrate the method and to discuss its advantages and limitations. When more training data become available the method can be used to make predictions elsewhere - probably first for limited areas and then globally.

Specific comments

Page 3 Section 2.1.2 (Predictors) Line 28 mentions that the continental crust was set at 1 billion years to represent significant older crust than that of the oceanic crust. If all the observed data (DSDP/ODP/IODP) are on oceanic crust, what is the importance/meaning of defining continental crust age?

AC: It is not true that all the data are from sites above oceanic crust. In fact, 142 of the 333 boreholes - 42% - were drilled on continental crust, e.g. in continental shelf regions. As the thermal regime of continental crust is different to that of oceanic crust - with old continental crust being of lower temperatures than young oceanic crust -, which affects density and hence p-wave velocity, we thought it reasonable to differentiate

between the two types of crust and their ages. CM: no changes made in the manuscript

Page 7 Section 3.3 (Predictor importance) Line 20 states that categorical predictors generally do not have any importance in prediction performance. Additionally, it is again discussed in the discussion section (Section 4.2- lines 8-14 page 9). What is the variance of your sampled data set in categorical predictors? For example, for a given test data set (i.e. fold) are all of your categorical predictors for that run a 1 or 0? If all of your test data set has only one categorical value then that predictor would be of no importance.

AC: We do not claim that categorical predictors "generally" do not have "any importance" in prediction performance. In the referenced line (now p. 8 line 10-11), we use the term "negligible importance", i.e. almost zero, and we explain that this only refers to the results of our own study, not to studies involving categorical predictors in general. The number of boreholes per predictor (for which the predictor is 1) varies between 2 (0.6%) and 191 (57%), on average, it is 42 (12.7%). We therefore agree that predictors with a very low representation will also be of low importance, and that this should be added as an explanation in the Discussion.

CM: We added the following sentence to the end of section 2.1 in the Methods: "Across the categorical predictors, the number of boreholes for which a predictor was set to 1 varied between 2 (0.6 %) and 191 (57.4 %); on average, the geological setting represented by a categorical predictor applied to 42 boreholes (12.7 %)."

We also included the sentence "The poor representation of some predictors, such as "cold_vent", "mud_volcano" and "hydroth_vent" in the dataset, causing these predictors to be 0 for all boreholes in some test folds, may likely explain the low importance of these predictors in the predictor ranking." in the last paragraph of section 4.2 in the Discussion (p. 10 lines 2-5).

Consider, if true, explicitly stating that predictions of this kind have not done with depth before. (page 2 ~ lines 16-20)

AC: As far as we know, predictions with depth have not been done before, and we agree that this should be stated in the text.

CM: We added a sentence to this paragraph (lines 21-23): "These studies were in general restricted to the prediction of one value per geographic location; the prediction of multiple values, such as depth profiles, has, to our knowledge, not been attempted before."

Minor suggestion to add in the abstract that this method is not designed to capture high variance in a p-wave velocity profile, but is instead intended to capture the overall trend of p-wave velocity profile.

AC: We agree that this should already be stated in the Abstract.

CM: We changed the sentence in line 9-10 (now lines 10-11) to read: "Here, we present a machine learning approach to predict the overall trend of seismic p-wave velocity (vp) as a function of depth (z) for any marine location."

It is stated and supported (Line 1 page 7; Figure 3) that the RFE CV 16 predictors prediction (green) is better than CV, max_features =22, 38 predictors however the error in the prediction is significantly higher for the green prediction with roughly the same % boreholes labelled as "good". Why do you consider green prediction to be so much better than yellow prediction? It might be useful if you explicitly state what your ultimate metric of correctness is (e.g. highest % correct or lowest error?)

AC: We did not mean to imply that one of the two runs provides better results than the other, and we also do not claim this anywhere. We merely stated the differences. However, we agree that this could have been easily misunderstood due to our ill use of the term "performance" - we meant performance to refer to both the highest % correct and the lowest error (i.e., in the same model), but we seem to have used it in other ways too, which must have been confusing. We now explain in more detail what we mean by performance and how our predictions were evaluated.

CM: Paragraph 3 in the Methods section 2.3 was changed to read "Performance of the RF model was evaluated in two ways: (1) by standard error metrics and (2) by the proportion of boreholes with predicted vp(z) superior to that of empirical functions. The standard error metrics root mean square error (RMSE), mean absolute error (MAE), and the coefficient of determination (R2) were calculated based on the comparison of the predicted and true vp(z) curves for each borehole in the test fold. RMSE, MAE and R2 of all test folds were then averaged to give final performance values." Throughout the manuscript, we also replaced the term "performance" where necessary, to make its use consistent. In the sentence reference above (now lines 19-22), we replaced "performance" by "prediction scores".

What is the final global spatial resolution? E.g. prediction of p-wave velocity profile every 1- degree, 5-min, etc.?

AC: We do not want to go so far as to give a final global spatial resolution for the prediction of vp. Our main aim was to investigate if it is at all possible to achieve realistic predictions of vp(z). We have shown that this is generally the case, however, our results also clearly indicate that more input data are required to overcome low prediction performance due to lack of suitable data. For this reason, we think that the prediction model needs to be improved further before a "final resolution" should be given. - In any case, one final resolution value likely would not be sufficient. Due to the heterogeneous depth distribution of the boreholes used (in addition to the heterogeneous spatial distribution), the resolution would vary with depth. Thus, separate resolution values would need to be determined for different depths (here: range 0-2500 m), which would likely be confusing and not very helpful for the reader.

CM: no changes made in the manuscript

Page 9 Section 4.2 (Most important predictors for the prediction of vp(z)) Lines 2-8 discuss how certain predictors are not used (porosity, density, pressure) as not all boreholes have depth associated measurements. However, some of the predictors used in

the prediction do not have a depth component (e.g. crustage). Applying this logic, why do you not use seafloor porosity (i.e. depositional porosity) or likewise predictors?

AC: We did not mean that we can only use predictors with depth measurements, we obviously also used depth-independent predictors. The point here (which was not well explained in the text) was that we could only use predictors that were available (or could be determined) for every borehole location. This did not apply to many of the e.g. porosity measurements, which had been measured in boreholes (with a depth component) but often not at the borehole locations at which vp had been measured. Even in the relatively few boreholes where both porosity (or density, pressure etc) and vp had been measured, the depth ranges did not always match - so there would have been depths with vp data but no porosity data. It was impossible to also account for such cases, which is why we decided to leave these parameters out. As reviewer 2 also asked for an explanation regarding choice of predictors, we clarified this in the Methods section 2.1.2. We also agree that a parameter like seafloor porosity, which is available as a global grid (we are assuming that the reviewer is referring to the grid by Martin et al., 2015), could easily have been added as a predictor. We did not do this at the time, and we hope the reviewer will understand that it is now too late to add new predictors to our study - as we state in section 4.2, there are several other predictors that could potentially be added, but this would have to be done in a future study. CM: We have clarified our choice of predictors by adding the following passage to the Methods section 2.1.2 (p. 4 lines 5-13): "... These predictors were parameters that were assumed to influence p-wave velocity. However, only predictors that could be obtained for each of the 333 borehole locations were used. Predictors such as latitude (lat), longitude (long), and water depth (wdepth) were taken from the borehole's metadata, whereas other predictors were extracted from freely available global datasets and grids (Table 1). In addition, predictors describing the borehole's geological setting were determined from the site descriptions given in the proceedings of each drilling campaign. Some parameters known to influence seismic velocity - e.g. porosity, density, or pressure - had to be left out as suitable datasets were not
available. Although some of these parameters had been measured in DSDP, ODP and IODP boreholes, they had not necessarily been logged at the same locations and depths at which vp data had been measured, and therefore could not be obtained at all of the 333 boreholes used."

No supplemental material was provided for the global prediction of p-wave velocity with depth. This paper should include the final global prediction of p-wave velocity with depth.

AC: No, we do not agree. As with the final spatial resolution, providing a final global prediction of vp at this stage (i.e. when the prediction model still requires optimization and is therefore not final yet) is neither feasible nor helpful. In fact, it would maybe give this method a bad reputation to deploy it prematurely. Furthermore, we show that one "final global prediction" would not be sufficient. We assume the reviewer expects a global map of final prediction values, similar to Fig. 4 in Taylor et al. (2019) or Fig. 1c in Martin et al. (2015). While such a map may be useful in cases with only one prediction value per location, in our case - taking into account the depth component of the predicted vp - a whole range of prediction maps would seem necessary, one for each depth. However, none of these maps would be of much use on its own. It would only show the variation of velocity at a certain depth, but we are interested in the variation (or trend) of velocity with depth (i.e., a profile), which is much better illustrated by the predicted vp(z) profiles (of which we show sufficient examples). Thus, we do not think a final global prediction is useful.

CM: no changes made in the manuscript

Technical corrections

Page 8 delete "the" on line 21: "by the at least 60% of test locations" CM: deleted "the" (now p. 9 line 8)

Page 8 line 3 consider changing "our results show that vp(z) profiles" to "our results

show that the general trend of vp(z) profiles" CM: We changed this sentence accordingly. (now line 24)

Page 16 Figure 2 caption (e) change "less good" to different word (substandard?) CM: We changed this to "lower-quality prediction".

Page 23 table 3, change words so they have consistent capitalization between table columns (e.g. Long and long) CM: We changed the capitalized letters accordingly (also in Table 2).

Page 12 Lee et al., 2019 citation is missing the publication year. AC: This paper was fully published just before we submitted our manuscript and we forgot to update the reference correctly. CM: Added publication year.

---

## Author Comment (AC2) · 9 Aug 2019

The manuscript, " Prediction of seismic p-wave velocity using machine learning", is a well-written description of a machine learning method -Random Forests - to predict seismic p-wave velocity as a function of depth for any a generic marine location. This manuscript is suitable for Copernicus, but the manuscript needs to be revised before it can be accepted. I have some suggestions here.

1 Introduction.

Page 2:

L24: You make the statement that the most widely used machine learning methods are ANNs, SVMs, and RFs. It is hard to convince people that these three algorithms are

the most widely used. For specific problems, some algorithms may be more common than the other algorithms. You may say that the most widely used machine learning includes ANNs, SVMs, and RFs.

AC: We agree with the reviewer that this is probably problem-dependent and should not be generalized here.

CM: We replaced "are" by "include". (now line 26)

L31: You mentioned that RF has been repeatedly found superior to other machine learning methods. You need to specify the particular problems that RF has been found superior to "other machine learning methods" in the text. And what other machine learning methods do you mean here? Please specify in the text.

AC: We have given more details on the particular studies and algorithms tested.

CM: We added 2 sentences after the first sentence of this paragraph (now p.3 lines 1-5): "For example, Li et al. (2011) tested 23 machine learning algorithms - including RF, SVM, and kriging methods - to predict mud content in marine sediments, and found that RF, along with RF combined with ordinary kriging or inverse distance squared, provided the best prediction results. Cracknell and Reading (2014) applied five machine learning methods to lithology classification of multispectral satellite data and reported higher classification accuracy for RF than for Naive Bayes, SVM, ANN, and k-Nearest Neighbors."

2 Methods

Page 3, section 2.1.2:

L25: How do you come up with these 38 predictors? Could you specify the reason why you choose these 38 predictors in this section?

AC: We agree that further information would be helpful in this section.

CM: We shortened the first sentence to "A total of 38 geological and spatial variables

were included as predictors (Table 1)." and added the following passage: "These predictors were parameters that were assumed to influence p-wave velocity. However, only predictors that could be obtained for each of the 333 borehole locations were used. Predictors such as latitude (lat), longitude (long), and water depth (wdepth) were taken from the borehole's metadata, whereas other predictors were extracted from freely available global datasets and grids (Table 1). In addition, predictors describing the borehole's geological setting were determined from the site descriptions given in the proceedings of each drilling campaign. Some parameters known to influence seismic velocity - e.g. porosity, density, or pressure - had to be left out as suitable datasets were not available. Although some of these parameters had been measured in DSDP, ODP and IODP boreholes, they had not necessarily been logged at the same locations at which vp data had been measured, and therefore could not be obtained at all of the 333 boreholes used."

Page 4, section 2.2:

L14: How do you define "performance"? I saw you mentioned performance in the later section 2.3. But it is better to define that when you first mention that.

AC: We agree that it is not always clear what we mean by the term "performance", and we also used it inconsistently to refer to the standard error metrics, the proportion of well predicted boreholes, or both - this is obviously confusing. By performance, we mean both the error metrics and the proportion of well predicted boreholes. We now explain this in paragraph 3 of section 2.3 (p. 5 lines 20-25) and removed/replaced the term in the previous paragraphs.

CM: Paragraph 3 was changed to read "Performance of the RF model was evaluated in two ways: (1) by standard error metrics and (2) by the proportion of boreholes with predicted vp(z) superior to that of empirical functions. The standard error metrics root mean square error (RMSE), mean absolute error (MAE), and the coefficient of determination (R2) were calculated based on the comparison of the predicted and true vp(z)

curves for each borehole in the test fold. RMSE, MAE and R2 of all test folds were then averaged to give final performance values." Throughout the manuscript, we replaced the term "performance" where necessary.

In addition, why do you choose 1000 trees? what is the maximum depth of each tree? How does the number of trees and depth affect the bias and variance of the prediction?

AC: Our study of relevant literature showed that most studies used either 500 or 1000 trees. In an early version of our prediction model, we ran RF repeatedly for numbers of trees between 2 and 1500 and evaluated model performance based on the OOB (out-of-bag) score. As the performance still varied after 500 trees but stabilized around 1000 trees, we chose 1000 trees. We did not repeat this procedure with our final prediction model (which no longer used the OOB approach), so it is possible that a lower number of trees might already have been sufficient. In that case, however, a higher number would not have decreased model performance. The depth of the trees was not defined and therefore not varied in the final prediction model. Early tests showed that performance was generally worse when maximum tree depths were specified (e.g. for max_depth = 5).

CM: no changes made in the manuscript

3 Results

Page 6, section 3.1:

The performance of an algorithm should be shown by both bias and variance. I only see the comparison of errors and percentage of boreholes with scores 2 and 3 in Fig. 3 and 4. How does the number of predictors and data smoothing affect the variance of the prediction?

AC: There is no strict rule that algorithm performance should always be evaluated by bias and variance. Many studies applying machine learning methods use other means to validate their results. We chose to evaluate performance by MAE, RMSE

and $R^2$, which have been used as performance measures by several other studies that predicted environmental parameters (e.g. Gasch et al. (2015), Ließ et al. (2016), Meyer et al. (2015, 2016)). Our own borehole percentage value serves as an additional measure. We think that our performance evaluation approach is now well described in section 2.3. The effects of varying numbers of predictors and data smoothing in terms of prediction performance are already described in the text.

CM: see above for changes regarding clarification of prediction performance

Since you only have 333 boreholes, 2% change due to different model runs only change scores of 7 boreholes. I am curious about the location distributions of those boreholes which changed their scores, and why their scores changed by changing the number of predictors or data smoothing.

AC: Unfortunately, our applied prediction method does not allow determining which boreholes changed their scores across different model runs. We agree that this would be an interesting aspect to look into, but in this case our model cannot easily be adapted accordingly, so this would likely require setting up a completely new model. This is beyond the scope of these revisions.

CM: no changes made in the manuscript

4 Discussion

Page 10:

L1-5: You made a strong statement about performance of RF. As I suggested in your introduction section, the performance of a machine learning algorithm really depends on situations.

AC: We agree that this can also be misunderstood to mean that RF is always the perfect choice, which is of course not the case. What we actually meant to say was that due to the issues with our dataset (spatially inhomogeneous, varying depth ranges, etc), it is much more likely that the cases of poor performance are due to the dataset itself, and

not due to the choice of machine learning algorithm.

CM: We rewrote the last sentence (now lines 31-33) to clarify this: "However, given the present dataset and its spatial inhomogeneity, we doubt that a different algorithm would lead to a significantly improved prediction performance for vp."

5 Conclusion

Page 10:

L15: RF is hard to extrapolate to data outside the range they have been seen. I doubted that RF can be used for geophysical modeling in areas lacking v_p(z) from boreholes or seismic data.

AC: This is why we recommend more data to be added - to increase the data ranges within the RF model and the likelihood that when the RF model is applied to new data, these data are within the ranges known to RF. We agree that at present, this is not always the case, which likely explains some of the lower-performing locations. However, we also point out that our RF model is not meant as a replacement for other sources of vp(z) data. It is meant only as an aid when no other means are available. We do not expect RF to ever replace or be superior (or even very close) to actual vp measurements or vp from seismic data (nor do we claim this in the manuscript). Our approach is only meant to provide an alternative to using an (unrealistic) constant velocity or empirically-derived vp(z) profiles, which are, as we show, often of lower quality than our predicted vp(z) profiles.

CM: no changes made in the manuscript

---

## Author Response (AR2)

GEOMAR | Wischhofstraße 1-3 | 24148 Kiel | Germany

Solid Earth
-editor in chief-

**Prof. Dr. Christian Berndt**
**Marine Geodynamics**

Tel  +49 431 600-2273
Fax  +49 431 600-2922
cberndt@geomar.de

+

**Final corrections SE-2019-58**                                    8. October 2019

Dear editor:

We have now uploaded the final version of the manuscript.

**GEOMAR**
Helmholtz-Zentrum für
Ozeanforschung Kiel

Wischhofstraße 1-3
24148 Kiel | Germany

Tel +49 431 600-0
Fax +49 431 600-2805
www.geomar.de

Deutsche Bank AG Kiel
BLZ 210 700 24
Kto. 144 8000

SWIFT/BIC DEUTDEDB210
IBAN DE 6921070024014480000

Steuernummer 2029745781
USt.-IdNr. DE281295378

+

We have changed the final sentence of the abstract according to the reviewer's suggestion.

Regarding her second comment: It is true that even if the same 10 folds are used in cross validation, variability in the model (and hence the results/errors) can be induced by randomly bootstrapped samples in the tree building process if the random seed is not set. However, we did set the random seed to a constant integer in all models. Therefore, all trees are the same in each RFE iteration. Consequently, variations between different models (e.g. the unscaled and scaled RFE cases) must be due to other reasons, which we discuss in the text. We have not changed anything in the text.

Yours sincerely,

Prof. Dr. Christian Berndt

**Stiftung des öffentlichen Rechts**
MinDir Volker Rieke, *Vorsitzender des Kuratoriums*
Prof. Dr. Peter Herzig, *Direktor* | Frank Spiekermann, *Verwaltungsdirektor*